# Efficient semidefinite-programming-based inference for binary and multi-class MRFs

**Chirag Pabbaraju**
Carnegie Mellon University
`cpabbara@cs.cmu.edu`

**Po-Wei Wang**
Carnegie Mellon University
`poweiw@cs.cmu.edu`

**J. Zico Kolter**
Carnegie Mellon University
Bosch Center for AI
`zkolter@cs.cmu.edu`

## Abstract

Probabilistic inference in pairwise Markov Random Fields (MRFs), i.e. computing the partition function or computing a MAP estimate of the variables, is a foundational problem in probabilistic graphical models. Semidefinite programming relaxations have long been a theoretically powerful tool for analyzing properties of probabilistic inference, but have not been practical owing to the high computational cost of typical solvers for solving the resulting SDPs. In this paper, we propose an efficient method for computing the partition function or MAP estimate in a pairwise MRF by instead exploiting a recently proposed coordinate-descent-based fast semidefinite solver. We also extend semidefinite relaxations from the typical binary MRF to the full multi-class setting, and develop a compact semidefinite relaxation that can again be solved efficiently using the solver. We show that the method substantially outperforms (both in terms of solution quality and speed) the existing state of the art in approximate inference, on benchmark problems drawn from previous work. We also show that our approach can scale to large MRF domains such as fully-connected pairwise CRF models used in computer vision.

## 1 Introduction

Undirected graphical models or Markov Random Fields (MRFs) are used in various real-world applications like computer vision, computational biology, etc. because of their ability to concisely represent associations amongst variables of interest. A general pairwise MRF over binary random variables $x \in \{-1, 1\}^n$ may be characterized by the following joint distribution

$$p(x) \propto \exp\left(x^T A x + h^T x\right), \qquad (1)$$

where $A \in \mathbb{R}^{n \times n}$ denotes the "coupling matrix" and encodes symmetric pairwise correlations, while $h \in \mathbb{R}^n$ consists of the biases for the variables. In this model, there are three fundamental problems of interest: (a) estimating the mode of the distribution, otherwise termed as *maximum a posteriori (MAP)* inference (b) estimating $p(x)$ for a configuration $x$ or generating samples from the distribution, and (c) learning the parameters $(A, h)$ given samples from the joint distribution. Since there are an exponential number of configurations in the support of the MRF, the problem of finding the true mode of the distribution is in general a hard problem. Similarly, to compute the probability $p(x)$ of any particular configuration, one has to compute the constant of proportionality in (1) which ensures that the distribution sums up to 1. This constant, denoted as $Z$, is called the *partition function*, where

$$Z = \sum_{x \in \{-1,1\}^n} \exp\left(x^T A x + h^T x\right).$$

Computing $Z$ exactly also involves summing up an exponential number of terms and is #P hard [17, 13] in general. The problem becomes harder still when we go beyond binary random variables and consider the case of a general multi-class MRF (also termed as a Potts model), where each

variable can take on values from a finite set. Since problem (b) above requires computing $Z$ accurately, several approximations have been proposed in the literature. These methods have typically suffered in the quality of their approximations in the case of problem instances where the entries of $A$ have large magnitude; this is referred to as the low-temperature setting [31, 29, 12, 26].

Recently, Park et al. [26] proposed a novel spectral algorithm that provably computes an approximate estimate of the partition function in time polynomial in the dimension $n$ and spectral properties of $A$. They show that their algorithm is fast, and significantly outperforms popular techniques used in approximate inference, particularly in the low-temperature setting. However, their experimental results suggest that there is still room for improvement in this setting. Furthermore, it is unclear how their method could be conveniently generalized to the richer domain of multi-class MRFs.

Another well-studied approach to compute the mode, i.e. the maximizer of the RHS in (1), is to relax the discrete optimization problem to a semidefinite program (SDP) [35, 1] and solve the SDP instead. Rounding techniques like randomized rounding [15] are then used to round the SDP solution to the original discrete space. In particular, Wang et al. [35] employ this approach in the case of a binary RBM and demonstrate impressive results. Frieze et al. [10] draw parallels between mode estimation in a general $k$-class Potts model and the MAX $k$-CUT problem, and suggest an SDP relaxation for the same. However, their relaxation has a *quadratic* number of constraints in the number of variables in the MRF. Therefore, using traditional convex program solvers employing the primal dual-interior point method [2] to solve the SDP would be computationally very expensive for large MRFs.

In this work, we propose solving a fundamentally different SDP relaxation for performing inference in a general $k$-class Potts model, that can be solved efficiently via a recently proposed low-rank SDP solver [32], and show that our method performs accurately and efficiently in practice, scaling successfully to large MRFs. Our SDP relaxation has only a *linear* number of constraints in the number of variables in the MRF. This allows us to exploit a low-rank solver based on coordinate descent, called the "Mixing method" [32], which converges extremely fast to a global solution of the proposed relaxation. We further propose a simple importance sampling-based method to estimate the partition function. Once we have solved the SDP, we state a rounding procedure to obtain samples in the discrete space. Since the rounding is applied to the optimal solution, the samples returned are closely clustered around the true mode in function value. Then, to ensure additional exploration in the space of the samples, we obtain a fraction of samples from the uniform distribution on the discrete hypercube. The combination results in an accurate estimate of the partition function.

Our experimental results show that our technique excels in both mode and partition function estimation, when compared to state-of-the-art methods like Spectral Approximate Inference [26], as well as specialized Markov Chain Monte Carlo (MCMC) techniques like Annealed Importance Sampling (AIS) [23], especially in the low temperature setting. Not only does our method outperform these methods in terms of accuracy, but it also runs significantly faster, particularly compared to AIS. We display these results on synthetic binary MRF settings drawn from Park et al. [26], as well as synthetic multi-class MRFs. Finally, we demonstrate that, owing to the efficiency of the fast SDP solver, our method is able to scale to large real-world MRFs used in image segmentation tasks.

## 2    Background and Related Work

**Variational methods and Continuous relaxations**    One popular class of approaches in approximate inference consists of framing a relevant optimization problem whose solution can be treated as a reasonable approximation to the true mode/partition function. This includes techniques employing the Gibbs variational principle [3] that solve an optimization problem over the set of all possible distributions on the random variables (generally intractable). Amongst these, the mean-field approximation [25], which makes the simplifying (and possibly inaccurate) assumption of a product distribution amongst the variables, is extremely popular. Belief propagation [37] is another popular algorithm used for inference that has connections to variational inference, but has strong theoretical guarantees only when the underlying MRFs are loop-free. In addition, several LP-based and SDP-based continuous relaxations [18, 30] to the discrete optimization problem have been proposed. In particular, Frieze et al. [10] model the problem of estimating the mode in a multi-class MRF as an instance of the MAX $k$-CUT problem and propose an SDP relaxation as well as rounding mechanism for the same. Generally, such SDP-based approaches are theoretically attractive to analyze [11, 27, 5, 24], but practically infeasible for large MRFs with many constraints due to their high computational cost.

**MCMC and Sampling-based methods**   Another class of approaches involves running MCMC chains whose stationary distribution is the one specified by (1). These methods run a particular number of MCMC steps and then do some sort of averaging over the samples at the end of the chain. Popular methods include Gibbs sampling [14] and Metropolis-Hastings [16]. A significant development in these methods was the introduction of annealing over a range of temperatures by Neal [23], which computes an unbiased estimate of the true partition function. Thereafter, several other methods in this line that employ some form of annealing and importance sampling have emerged [4, 21, 6]. However, it is typically difficult to determine the number of steps that the MCMC chain requires to converge to the stationary distribution (denoted mixing time). Further, as mentioned above, both variational methods (due to their propensity to converge to suboptimal solutions) and MCMC methods (due to large mixing times) are known to underperform in the low-temperature setting.

**Other methods**   Some other popular techniques for inference include variable elimination methods like bucket elimination [7, 8] that typically use a form of dynamic programming (DP) to approximately marginalize the variables in the model one-by-one. A significant recent development in this line, which is also based on DP, is the spectral approach by Park et al. [26]. By viewing all possible configurations of the random variables in the function space, Park et al. [26] build a bottom-up approximate DP tree which yields a fully-polynomial time approximation scheme for estimating $Z$, and markedly outperforms other standard techniques. However, as mentioned above, it is a priori unclear how their method could be extended to the multi-class Potts model, since their bottom-up dynamic programming chain depends on the variables being binary-valued. Other approximate algorithms for estimating the partition function with theoretical guarantees include those that employ discrete integration by hashing [9] as well as quadrature-based methods [28]. While our method does not purely belong to either of the two categories mentioned above (since it involves *both* an SDP relaxation and importance sampling), it successfully generalizes to multi-class MRFs.

## 3   Estimation of the mode in a general $k$-class MRF

In this section, we formulate the SDP relaxation which we propose to solve for mode estimation in a $k$-class Potts model. First, we state the optimization problem for mode estimation in a binary MRF:

$$\max_{x \in \{-1,1\}^n} \quad x^T A x + h^T x. \tag{2}$$

We observe that the above problem can be equivalently stated as below:

$$\max_{x \in \{-1,1\}^n} \sum_{i=1}^n \sum_{j=1}^n A_{ij} \hat{\delta}(x_i, x_j) + \sum_{i=1}^n \sum_{l \in \{-1,1\}} \hat{h}_i^{(l)} \hat{\delta}(x_i, l); \quad \text{where} \quad \hat{\delta}(a, b) = \begin{cases} 1 & \text{if } a = b \\ -1 & \text{otherwise.} \end{cases} \tag{3}$$

The equivalence in (2) and (3) is readily achieved by setting $\hat{h}_i^{(l)}$ such that $h_i = \hat{h}_i^{(1)} - \hat{h}_i^{(-1)}$. However, viewing the optimization problem thus helps us naturally extend the problem to general $k$-class MRFs where the random variables $x_i$ can take values in a discrete domain $\{1, \ldots, k\}$ (denoted $[k]$). For the general case, we can frame a discrete optimization problem as follows:

$$\max_{x \in [k]^n} \sum_{i=1}^n \sum_{j=1}^n A_{ij} \hat{\delta}(x_i, x_j) + \sum_{i=1}^n \sum_{l=1}^k \hat{h}_i^{(l)} \hat{\delta}(x_i, l). \tag{4}$$

where we are now provided with bias vectors $\hat{h}^{(l)}$ for each of the $k$ classes.

**Efficient SDP relaxation**   We now derive an SDP-based relaxation to (4) that grows only *linearly* in $n$. To motivate this approach, we note that for the case of (4) (without the bias terms), Frieze et al. [10] first state an equivalent optimization problem defined over a simplex in $\mathbb{R}^{k-1}$, and go on to derive the following relaxation for which theoretical guarantees hold:

$$\max_{v_i \in \mathbb{R}^n, \ \|v_i\|_2 = 1 \ \forall i \in [n]} \sum_{i=1}^n \sum_{j=1}^n A_{ij} v_i^T v_j$$

$$\text{subject to} \quad v_i^T v_j \geq -\frac{1}{k-1} \quad \forall i \neq j. \tag{5}$$

The above problem (5) can be equivalently posed as a convex problem over PSD matrices $Y$, albeit with $\sim n^2$ entry-wise constraints $Y_{ij} \geq -1/k-1$ corresponding to each pairwise constraint in $v_i, v_j$. Thus, for large $n$, solving (5) via traditional convex program solvers would be very expensive. Note further that, unlike the binary case (where the pairwise constraints hold trivially), it would also be challenging to solve this problem with low-rank methods, due to the quadratic number of constraints.

Towards this, we propose an alternate relaxation to (4) that reduces the number of constraints to be linear in $n$. Observe that the pairwise constraints in (5) are controlling the separation between $v_i, v_j$ and trying to keep them roughly aligned with the vertices of a simplex. With this insight, we try to incorporate the functionality of these constraints within the criterion by plugging them in as part of the bias terms. Specifically, let us fix $r_1, \ldots, r_k \in \mathbb{R}^n$ on the vertices of a simplex, so that

$$r_l^T r_{l'} = \begin{cases} 1 & \text{if } l = l' \\ -\frac{1}{k-1} & \text{if } l \neq l'. \end{cases}$$

Then, we can observe that the following holds:

$$\hat{\delta}(x_i, x_j) = \frac{2}{k}\left((k-1)r_{x_i}^T r_{x_j} + 1\right) - 1, \tag{6}$$

so that solving the following discrete optimization problem is identical to solving (4):

$$\max_{v_i \in \{r_1, \ldots, r_k\} \ \forall i \in [n]} \sum_{i=1}^{n} \sum_{j=1}^{n} A_{ij} v_i^T v_j + \sum_{i=1}^{n} \sum_{l=1}^{k} \hat{h}_i^{(l)} v_i^T r_l. \tag{7}$$

The motivation here is that we are trying to mimic the $\hat{\delta}$ operator in (4) via inner products, but in such a way that the bias coefficients $\hat{h}_i^{(l)}$ determine the degree to which $v_i$ is aligned with a particular $r_l$. Thus, intuitively at least, we have incorporated the pairwise constraints in (5) within the criterion. As the next step, we simply relax the domain of optimization in (7) from the discrete set $\{r_i\}_{i=1}^{k}$ to unit vectors in $\mathbb{R}^n$ so as to derive the following relaxation:

$$\max_{v_i \in \mathbb{R}^n, \ \|v_i\|_2 = 1 \ \forall i \in [n]} \sum_{i=1}^{n} \sum_{j=1}^{n} A_{ij} v_i^T v_j + \sum_{i=1}^{n} v_i^T \sum_{l=1}^{k} \hat{h}_i^{(l)} r_l. \tag{8}$$

Now, let $H \in \mathbb{R}^{n \times k}$ such that $H_{ij} = \hat{h}_i^{(j)}$. Define the block matrix $C \in \mathbb{R}^{(k+n) \times (k+n)}$ such that:

$$C = \begin{bmatrix} 0 & \frac{1}{2} \cdot H^T \\ \frac{1}{2} \cdot H & A \end{bmatrix}.$$

Then, the following convex program is equivalent to (8):

$$\max_{Y \succeq 0} \ Y \cdot C$$

$$\text{subject to} \ \ Y_{ii} = 1 \ \forall i \in [k+n]; \quad Y_{ij} = -\frac{1}{k-1} \ \ \forall i \in [k], \ i < j \leq k. \tag{9}$$

We defer the proof of the equivalence between (8) and (9) to Appendix A. Note that the number of constraints in (9) is now only $n + k + \frac{k(k-1)}{2} = n + \frac{k(k+1)}{2}$ i.e. *linear* in $n$ as opposed to the quadratic number of constraints in (5). We can then use the results by Barvinok [1] and Pataki [27], which state that there indeed exists a low-rank solution to (9) with rank $d$ at most $\left\lceil \sqrt{2\left(n + \frac{k(k+1)}{2}\right)} \right\rceil$. Thus, we can instead work in the space $\mathbb{R}^d$, leading to the following optimization problem:

$$\max_{v_i \in \mathbb{R}^d, \ \|v_i\|_2 = 1 \ \forall i \in [n]} \sum_{i=1}^{n} \sum_{j=1}^{n} A_{ij} v_i^T v_j + \sum_{i=1}^{n} v_i^T \sum_{l=1}^{k} \hat{h}_i^{(l)} r_l. \tag{10}$$

This low-rank relaxation can then directly be solved in its existing non-convex form by using the method for solving norm-constrained SDPs by Wang et al. [32] called the "mixing method", which we refer to as $M^4$ ("Mixing Method for Multi-class MRFs"). $M^4$ derives closed-form coordinate-descent updates for the maximization in (10), and has been shown [34, 33] to reach accurate solutions in just a few iterations. Pseudocode for solving (10) via $M^4$ is given in the block for Algorithm 1.

| **Algorithm 1** Solving (10) via $M^4$ | **Algorithm 2** Rounding in the multi-class case |
|---|---|
| **Input:** $A, \{v_i\}_{i=1}^n, \{\hat{h}^{(l)}\}_{l=1}^k, \{r_l\}_{l=1}^k$ | **Input:** $\{v_i\}_{i=1}^n, \{r_l\}_{l=1}^k$ |
| 1: **procedure** $M^4$: | 1: **procedure** ROUNDING: |
| 2:    Initialize $num\_iters$ | 2:    Sample $\{m_l\}_{l=1}^k \sim \text{Unif}(\mathcal{S}^d)$ |
| 3:    **for** $iter = 1, 2 \ldots, num\_iters$ **do** | 3:    **for** $i = 1, 2 \ldots, n$ **do** |
| 4:        **for** $i = 1, 2 \ldots, n$ **do** | 4:        $x_i \leftarrow \arg\max_{l \in [k]} \; v_i^T m_l$ |
| 5:          $g_i \leftarrow 2 \sum_{j \neq i} A_{ij} v_j + \sum_{l=1}^k \hat{h}_i^{(l)} r_l$ | 5:    **end for** |
| 6:          $v_i \leftarrow \frac{g_i}{\|g_i\|_2}$ | 6:    **for** $i = 1, 2 \ldots, n$ **do** |
| 7:        **end for** | 7:        $x_i \leftarrow \arg\max_{l \in [k]} \; m_{x_i}^T r_l$ |
| 8:    **end for** | 8:    **end for** |
| 9:    **return** $v_1, \ldots, v_n$ | 9:    **return** $x$ |
| 10: **end procedure** | 10: **end procedure** |

Once we have a solution $v_1, \ldots, v_n$ to (10), we still require a technique to round back these vectors to configurations in the discrete space $[k]^n$. For this purpose, Frieze et al. [10] propose a natural extension to the technique of randomized rounding suggested by Goemans et al. [15], which involves rounding the $v_i$s to $k$ randomly drawn unit vectors. We further extend their approach as described in Algorithm 2 for the purposes of rounding our SDP relaxation (10). In the first step in Algorithm 2, we sample $k$ unit vectors $\{m_l\}_{l=1}^k$ uniformly on the unit sphere $\mathcal{S}^d$ and perform rounding as in Frieze et al. [10]. However, we need to reconcile this rounding with the truth vectors on the simplex. Thus, in the second step, we reassign each rounded value to a truth vector: if $v_i$ was mapped to $m_l$ in the first step, we now map it to $r_{l'}$ such that $m_l$ is closest to $r_{l'}$. In this way, we can obtain a bunch of rounded configurations, and output as the mode the one that has the maximum criterion value in (4).

**Alternate relaxation to (7)**    $M^4$ provably solves the optimization problem in (10), but the rounded solution after applying Algorithm 2 lacks any approximation guarantees. This is because we simply discarded the pairwise constraints which existed in the original formulation (5) of Frieze et al. [10]. In order to remedy this, we further propose an alternate relaxation, so as to obtain an approximation ratio for the rounded solution.

To ensure that $v_i, v_j$ satisfy the pairwise constraints in 5, we adopt an alternate parameterization of the $v_i$s in terms of auxiliary variables $z_i \in \mathbb{R}^d$ for $d = m \cdot k, m \in \mathbb{Z}$ (we want to be able to segment each $z_i$ into $k$ blocks). Let $C = \frac{k}{k-1} \left( I_d - \frac{1}{k} \left( 1_{k \times k} \otimes I_m \right) \right)$ where $1_{k \times k}$ is filled with 1s, and let $C = S^T S$ denote the Cholesky decomposition of $C$. Further, let $z_i^b \in \mathbb{R}^m$ denote the $b^{th}$ block in $z_i$. Then, we frame the following optimization problem:

$$\max_{z_i \in \mathbb{R}^d \; \forall i \in [n]} \sum_{i=1}^n \sum_{j=1}^n A_{ij} v_i^T v_j + \sum_{i=1}^n v_i^T \sum_{l=1}^k \hat{h}_i^{(l)} r_l$$

$$\text{subject to} \quad z_i \geq 0, \quad \left\| \sum_{b=1}^k z_i^b \right\|_2^2 = 1, \quad v_i = S z_i \;\; \forall i \in [n] \tag{11}$$

We defer the detailed derivation for formulating (11) and solving it by Algorithm 3 (denoted as $M^4$+) to Appendix B. Here, we simply describe the motivation for such a parameterization of the $v_i$s. The approximation guarantees of Frieze et al. [10] hold for any set of $v_i$s lying in the feasible set of (5). Concretely, for $\|v_i\|_2 = 1$ and $v_i^T v_j \geq -1/k - 1$, we have that $\mathbb{E}[f(\text{Rounding}(V))] \geq \alpha \cdot f(V)$, where we denote the objective by $f$ and $V = \{v_1, \ldots, v_n\}$. Thus, if we find a set of feasible $v_i$s such that $f(V) \geq f_{discrete}^\star$, where $f_{discrete}^\star$ refers to the solution to (7), we have the required guarantee on the expected value of the rounding. In the above, the parameterization of the $v_i$s via the $z_i$s and $S$, together with the positivity constraints on $z_i$, *ensure* that $v_i, v_j$ satisfy the pairwise constraints. In addition, step 10 in Algorithm 3 ensures that after each round of updates, $\|z_i\|_2 = 1$ and consequently, $\|v_i\|_2 = 1$ for all $i$. Thus, we have that the $v_i$s after each round of updates in $M^4$+ lie in the required feasible set. Further, we empirically observe that at convergence of Algorithm 3, $f(V) > f_{discrete}^\star$ always (in fact, the solution is within 5% of the true solution to (11)). To summarize, provided that Algorithm 3 converges to a set of $v_i$s such that $f(V) > f_{discrete}^\star$, we have the approximation guarantees of Frieze et al. [10] for the rounded solution.

**Algorithm 3** Solving (11) via $M^4+$

**Input:** $A, \{z_i\}_{i=1}^n, \{\hat{h}^{(l)}, r_l\}_{l=1}^k, C = S^T S$

1: **procedure** $M^4+$:
2:     Initialize $num\_iters$
3:     **for** $iter = 1, 2 \ldots, num\_iters$ **do**
4:         **for** $i = 1, 2 \ldots, n$ **do**
5:             $g \leftarrow 2 \sum\limits_{j \neq i}^n A_{ij} C z_j + S^T \sum\limits_{l=1}^k \hat{h}_i^{(l)} r_l$
6:             **for** $j = 1, 2 \ldots, m$ **do**
7:                 Pick any $b(j) \in \arg\max_b g_j^b$
8:                 $g_j^b \leftarrow \begin{cases} (g_j^b)_+ & \text{if } b = b(j) \\ 0 & \text{otherwise} \end{cases}$
9:             **end for**
10:             $z_i \leftarrow \frac{g}{\|g\|_2}$
11:         **end for**
12:     **end for**
13:     **return** $S z_1, \ldots, S z_n$
14: **end procedure**

**Algorithm 4** Estimation of $Z$

**Input:** $k, \{v_i\}_{i=1}^n, \{r_l\}_{l=1}^k$

1: **procedure** PARTITIONFUNCTION:
2:     Initialize $R \in \mathbb{Z}, X_{p_v} = \{\}, X_\Omega = [\,]$
3:     **for** $i = 1, 2 \ldots, R$ **do**
4:         Sample $x \sim p_v$ using Algorithm 2
5:         If $x$ not in $X_{p_v}$, add $x$ to $X_{p_v}$
6:     **end for**
7:     $q \leftarrow \frac{1}{k^n - |X_{p_v}|}$
8:     **for** $i = 1, 2 \ldots, R$ **do**
9:         Sample $x \sim Unif([k]^n \setminus X_{p_v})$
10:         Append $x$ to $X_\Omega$
11:     **end for**
12:     $\hat{Z} \leftarrow \sum\limits_{x \in X_{p_v}} e^{f(x)} + \frac{1}{R} \sum\limits_{x \in X_\Omega} \frac{e^{f(x)}}{q}$
13:     **return** $\hat{Z}$
14: **end procedure**

## 4 Estimation of the partition function

In this section, we deal with the other fundamental problem in inference: estimating the partition function. Following Section 3 above, the joint distribution in a $k$-class MRF can be expressed as:

$$p(x) \propto \exp\left(\underbrace{\sum_{i=1}^n \sum_{j=1}^n A_{ij} \hat{\delta}(x_i, x_j) + \sum_{i=1}^n \sum_{l=1}^k \hat{h}_i^{(l)} \hat{\delta}(x_i, l)}_{f(x)}\right). \tag{12}$$

As stated previously, the central aspect in computing the probability of a configuration $x$ in this model is being able to calculate the partition function $Z$. The expression for the partition function in (12) is:

$$Z = \sum_{x \in [k]^n} \exp\left(\sum_{i=1}^n \sum_{j=1}^n A_{ij} \hat{\delta}(x_i, x_j) + \sum_{i=1}^n \sum_{l=1}^k \hat{h}_i^{(l)} \hat{\delta}(x_i, l)\right). \tag{13}$$

We begin with the intuition that the solution to (4) can be useful in computing the partition function. This intuition indeed holds if the terms in the summation are dominated by a few terms with large magnitude, which happens to be the case when $A$ has entries with large magnitude (low temperature setting). The hope then is that the rounding procedure described above more often than not rounds to configurations that have large $f$ values (ideally close to the mode). In that case, a few iterations of rounding would essentially yield all the configurations that make up the entire probability mass (a phenomenon which we empirically confirm ahead).

With this motivation, we describe a simple algorithm to estimate $Z$ that exploits the solution to our proposed relaxations. The rounding procedure described in Algorithm 2 induces a distribution on $x$ in the original space. Let us denote this distribution as $p_v$. For the case when $d = 2$, Wang et al. [35] propose a geometric technique for exactly calculating $p_v$, and derive an importance sampling estimate of $Z$ based on the empirical expectation $\hat{\mathbb{E}}\left[\exp(f(x))/p_v(x)\right]$. However, this approach does not scale to higher dimensions. Further, note that for small values of $d$, $p_v$ does not have a full support of $[k]^n$. Thus, an importance sampling estimate computed solely on $p_v$ would not be theoretically unbiased. Consequently, we propose using Algorithm 4 to estimate $Z$. First, we do a round of sampling from $p_v$, and store all the unique $x$'s seen in $X_{p_v}$. At this point, the hope is that $X_{p_v}$ stores all the $x$'s that constitute a bulk of the probability mass. Thereafter, to encourage exploration and ensure that our sampling process has a full support of $[k]^n$, we do a round of importance sampling from the uniform distribution on $[k]^n \setminus X_{p_v}$ and combine the result with the samples stored in $X_{p_v}$. For the estimate of $Z$ thus obtained, the following guarantee can be easily shown (refer to Appendix C for proof):

**Theorem 1.** *The estimate $\hat{Z}$ given by Algorithm 4 is unbiased i.e. $\mathbb{E}[\hat{Z}] = Z$.*

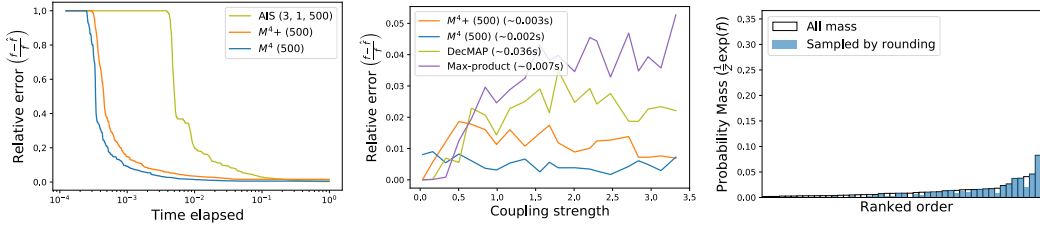

(a) Complete graph, $k = 5, n = 7$    (b) Complete graph $k = 5, n = 7$    (c) Mass sampled $k = 5, n = 7$

Figure 1: (a) Mode estimation - comparison with AIS (x-axis on log-scale) (b) Mode estimate comparison w/ max-product BP and decimation (c) Randomized rounding samples most of the mass

## 5 Experimental Results

In this section, we validate our formulations on a variety of MRF settings, both synthetic and real-world. Following its usage in Park et al. [26], we first state the notion of "coupling strength" of a matrix $A \in \mathbb{R}^{n \times n}$, which determines the temperature of the problem instance:

$$CS(A) = \frac{1}{n(n-1)} \sum_{i \neq j} |A_{ij}|. \tag{14}$$

As in the setup in Park et al. [26], the coupling matrices are generated as follows: for a coupling strength $c$, the entries on edges in $A$ are sampled uniformly from $[-c', c']$, where $c'$ is appropriately scaled so that $CS(A) \approx c$. The biases are sampled uniformly from $[-1, 1]$. We generate random complete graphs and Erdös-Renyi (ER) graphs. While generating ER graphs, we sample an edge in $A$ with probability $0.5$. We perform experiments on estimating the mode (Section 3) as well as $Z$ (Section 4). The algorithms we mainly compare to in our experiments are AIS [23], Spectral Approximate Inference (Spectral AI) [26] and the method suggested by Wang et al. [35]. We note that for the binary MRFs considered in the partition function task, Park et al. [26] demonstrate that they significantly outperform popular algorithms like belief propagation, mean-field approximation and mini-bucket variable elimination (Figures 3(a), 3(c) in Park et al. [26]). Hence, we simply compare to Spectral AI. For AIS, we have 3 main parameters: $(K, num\_cycles, num\_samples)$. We provide a description of these parameters along with complete pseudocode of our implementation of AIS in Appendix D. All the results in any synthetic setting are averaged over 100 random problem instances.[1]

### 5.1 Mode estimation

We compare the quality of the mode estimate over progress of our methods (rounding applied to $M^4$ and $M^4+$) and AIS. On the x-axis, we plot the time elapsed for either method, and on the y-axis, we plot the relative error $\frac{f - \hat{f}}{f}$. Figure 1a shows the comparison for $k = 5$ and $CS(A) = 2.5$. In the legend, the number in parentheses following our methods is the number of rounding iterations, while those following AIS are $(K, num\_cycles, num\_samples)$ respectively. We observe from the plots that our methods are able to achieve a near optimal mode much quicker than AIS, underlining the efficacy of our method. Next, we compare the quality of the mode estimates given by both of our relaxations with the max-product belief propagation and decimation (DecMAP) algorithms provided in libDAI [22]. Across a range of coupling strengths, we plot the relative error of the mode estimates given by each method, for $k = 5$. We can observe (Figure 1b) that both our relaxations provide mode estimates that have very small relative error ($\sim 0.018$ at worst), whilst also being faster. Additional plots comparing our methods as well as timing experiments are provided in Appendix E.

### 5.2 Partition function estimation

We now evaluate the accuracy of the partition function estimate given by our Algorithm 4 applied on the $M^4$ solution. First, we empirically verify our intuition about randomized rounding returning configurations that account for most of the probability mass in (12). For a 5-class MRF with

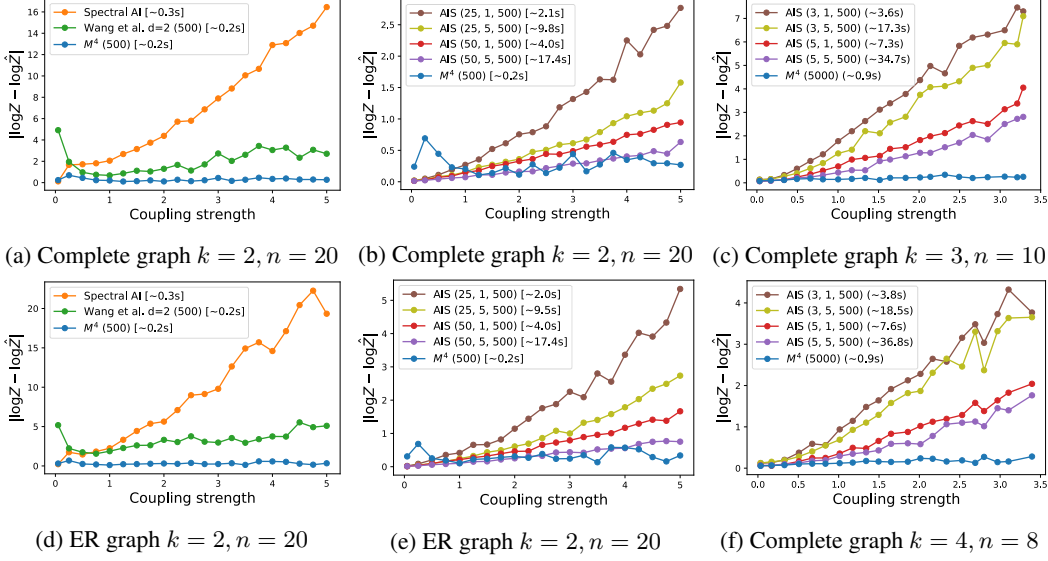

Figure 2: Estimation of $Z$

$CS(A) = 2.5$, we bucket the attainable $f$ values over different configurations of $x$, and compute the probability mass in each bucket. The buckets are then arranged in an increasing order of the probability mass, and the bars in Figure 1c show the mass in this order. Then, we obtain 1000 samples via randomized rounding, and fill in with blue the probability mass corresponding to the observed samples. We can observe that with just 1000 iterations of rounding, the obtained samples constitute most of the probability mass. Next, we consider coupling matrices over a range of coupling strengths and plot the error $|\log Z - \log \hat{Z}|$ against the coupling strength. We note here that for $k > 2$, there is no straightforward way in which we could extend the formulation in Spectral AI [26] to multiple classes; hence we only provide comparisons with AIS in this case. In the binary case (Figures 2a, 2d), we can observe that our estimates are more accurate than both Spectral AI [26] and Wang et al. [35] almost everywhere. Importantly, in the high-coupling strength setting, where the performance of Spectral AI [26] becomes pretty inaccurate, we are still able to maintain high accuracy. We also note that with just 500 rounding iterations, the running time of our algorithm is faster than Spectral AI [26]. We also provide comprehensive comparisons with AIS in Figures 2b, 2e, 2c, 2f, 3a over a range of parameter settings of $K$ and $num\_cycles$. We can see in the plots that on increasing the number of temperatures ($K$), the AIS estimates become more accurate, but suffer a lot with respect to time. Finally, we also analyze the performance of Algorithm 4 applied to the $M^4$+ solution in Figures 3b, 3c, 3d. We observe that the $M^4$+ estimates for $Z$ are slightly worse when compared to $M^4$ for larger $k$, but still much more accurate and efficient when compared to AIS.

### 5.3 Image segmentation

In this section, we demonstrate that our method of inference is able to scale up to large fully connected CRFs used in image segmentation tasks. Here, we consider the setting as in DenseCRF [19] where the task is to compute the configuration of labels $x \in [k]^n$ for the pixels in an image that maximizes:

$$\max_{x \in [k]^n} \sum_{i<j} \mu(x_i, x_j) \bar{K}(f_i, f_j) + \sum_i \psi_u(x_i).$$

The first term provides pairwise potentials where $\bar{K}$ is modelled as a Gaussian kernel that measures similarity between pixel-feature vectors $f_i, f_j$ and $\mu$ is the label compatibility function. The second term corresponds to unary potentials for individual pixels. As in the SDP relaxation described above, we relax each pixel to a unit vector in $\mathbb{R}^d$. We model $\mu$ via an inner product, and base the unary potentials $\phi_u$ on rough annotations provided with the images to derive the following objective:

$$\max_{v_i \in \mathbb{R}^d, \, \|v_i\|_2 = 1 \, \forall i \in [n]} \sum_{i<j} \bar{K}(f_i, f_j) v_i^T v_j + \theta \sum_{i=1}^{n} \sum_{l=1}^{k} \log p_{i,l} \cdot v_i^T r_l. \tag{15}$$

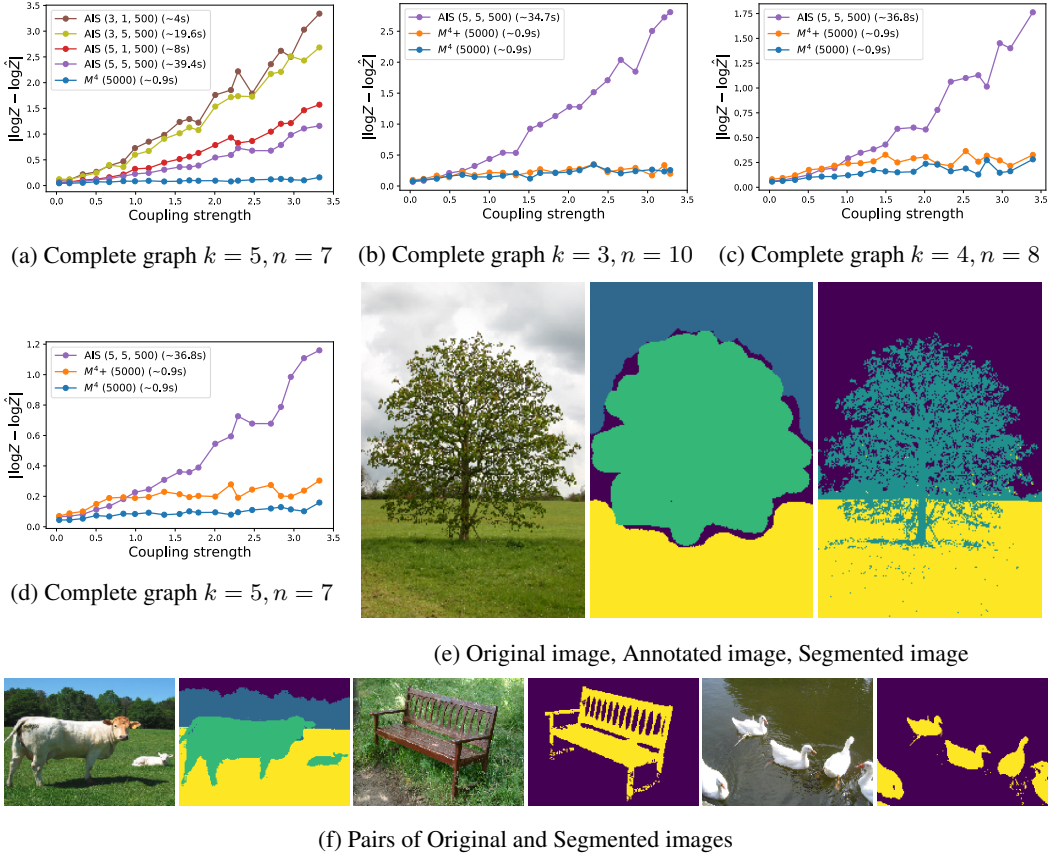

(a) Complete graph $k = 5, n = 7$    (b) Complete graph $k = 3, n = 10$    (c) Complete graph $k = 4, n = 8$

(d) Complete graph $k = 5, n = 7$

(e) Original image, Annotated image, Segmented image

(f) Pairs of Original and Segmented images

Figure 3: (a), (b), (c), (d) Estimation of $Z$ (e) For the tree, we show the original image, annotations and segmented image (f) Segmentations computed on other images based on similar annotations

In the second term above, $\log p_{i,l}$ plugs in our prior belief based on annotations for the $i^{th}$ pixel being assigned the $l^{th}$ label. The coefficient $\theta$ helps control the relative weight on pairwise and unary potentials. We note here that running MCMC-based methods on MRFs with as many nodes as pixels in standard images is generally infeasible. However, we can solve (15) efficiently via the mixing method. At convergence, using the rounding scheme described in Algorithm 2, we are able to obtain accurate segmentations of images (Figures 3e, 3f), competitive with the quality presented in DenseCRF [19]. More details regarding the setting here are described in Appendix G.

## 6 Conclusion and Future Work

In this paper, we presented a novel relaxation to estimate the mode in a general $k$-class Potts model that can be written as a low-rank SDP and solved efficiently by a recently proposed low-rank solver based on coordinate descent. We further introduced a relaxation that allows for approximation guarantees. We also proposed a simple and intuitive algorithm based on importance sampling which guarantees an unbiased estimate of the partition function. We set up experiments to empirically study the performance of our method as compared to relevant state-of-the-art methods in approximate inference, and verified that our relaxation provides an accurate estimate of the mode, while our algorithm for computing the partition function also gives fast and accurate estimates. We also demonstrated that our method is able to scale up to very large MRFs in an efficient manner.

The simplicity of our algorithm also lends itself to certain areas for improvement. Specifically, in the case of MRFs that have many well-separated modes, an accurate estimate of $Z$ should require sampling around each of the modes. Although we did empirically observe that randomized rounding samples most of the probability mass, the next steps involve studying other structured sampling mechanisms that indeed guarantee adequate sampling around each of the modes.

## Broader Impact

Probabilistic inference has been used in a number of domains including, e.g. the image segmentation domains highlighted in our final experimental results section. However, the methods have also been applied extensively to biological applications, such as a protein side chain prediction or protein design [36]. Such applications all have the ability to be directly affected by upstream algorithmic improvements to approximate inference methods. This also, however, applies to potentially questionable applications of machine learning, such as those used by automated surveillance systems. While it may be difficult to assess the precise impact of this work in such domains (especially since the vast majority of deployed systems are based upon deep learning methods rather than probabilistic inference at this point), these are applications that should be considered in the further development of probabilistic approaches.

From a more algorithmic perspective, many applications of approximate inference in recent years have become dominated by end-to-end deep learning approaches, forgoing application of probabilistic inference altogether. One potential advantage of our approach, which we have not explored in this current work, is that because it is based upon a continuous relaxation, the probabilistic inference method we present here can itself be made differentiable, and used within an end-to-end pipeline. This has potentially potentially positive effects (it could help in the interpretability of deep networks, for example), but also negative effects, such as the possibility that the inference procedure itself actually becomes *less* intuitively understandable if it's trained solely in an end-to-end fashion. We hope that both these perspectives, as well as potential enabled applications, are considered in the possible extension of this work to these settings.

## Acknowledgments

Po-Wei Wang is supported by a grant from the Bosch Center for Artificial Intelligence.

## Footnotes

[1]Source code for our experiments is available at https://github.com/locuslab/sdp_mrf

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
