[Supplementary Material]

# A Proof of Equivalence of (8) and (9)

We state (8) and (9) again. Let the vectors $r_1, \ldots, r_k \in \mathbb{R}^n$ be fixed on the simplex. Then, (8) is stated as follows:

$$\max_{v_i \in \mathbb{R}^n, \; \|v_i\|_2 = 1 \; \forall i \in [n]} \sum_{i=1}^{n} \sum_{j=1}^{n} A_{ij} v_i^T v_j + \sum_{i=1}^{n} v_i^T \sum_{l=1}^{k} \hat{h}_i^{(l)} r_l. \tag{8}$$

Let $H \in \mathbb{R}^{n \times k}$ such that $H_{ij} = \hat{h}_i^{(j)}$. Define the block matrix $C \in \mathbb{R}^{(k+n) \times (k+n)}$ such that:

$$C = \begin{bmatrix} 0 & \frac{1}{2} \cdot H^T \\ \frac{1}{2} \cdot H & A \end{bmatrix}.$$

Then, (9) is stated as follows:

$$\max_{Y \succeq 0} \; Y \cdot C \tag{9}$$
$$\text{subject to} \quad Y_{ii} = 1 \; \forall i \in [n+k]$$
$$Y_{ij} = -\frac{1}{k-1} \quad \forall i \in [k], \; i < j \leq k.$$

We will show that the optimal solutions to both these optimization problems are equal. Consider any $v_1, \ldots, v_n \in \mathbb{R}^n$ in the feasible set of (8). Then, corresponding to these $v_1, \ldots, v_n$, consider the matrix $Y \in \mathbb{R}^{(k+n) \times (k+n)}$ defined as follows:

$$Y = \begin{bmatrix} r_1^T \\ \vdots \\ r_k^T \\ v_1^T \\ \vdots \\ v_n^T \end{bmatrix} \begin{bmatrix} r_1 & \cdots & r_k & v_1 & \cdots & v_n \end{bmatrix}.$$

Clearly, $Y \succeq 0$. Further, since $r_1, \ldots, r_k$ are on the simplex and $v_1, \ldots, v_n$ are in the feasible set of (8), this matrix $Y$ satisfies the constraints in (9). Thus, $Y$ lies in the feasible set of (9). Also, because of the way in which the block matrix $C$ is defined, we can verify that:

$$Y \cdot C = \sum_{i=1}^{n} \sum_{j=1}^{n} A_{ij} v_i^T v_j + \sum_{i=1}^{n} v_i^T \sum_{l=1}^{k} \hat{h}_i^{(l)} r_l.$$

Thus, for any $v_1, \ldots, v_n$ in the feasible set of (8), we have a corresponding $Y$ in the feasible set of (9) such that the criterion values match.

Now, consider any $Y$ in the feasible set of (9). Since $Y \succeq 0$, we can compute its Cholesky decomposition as $Y = U^T U$ for some $U \in \mathbb{R}^{(k+n) \times (k+n)}$. Denote the first $k$ columns in $U$ as $r_1', \ldots, r_k'$ and the last $n$ columns of $U$ as $v_1', \ldots, v_n'$. Then, since $Y$ satisfies the constraints in (9), we have that $\|v_i'\|_2 = 1$ for all $i \in [n]$. Also, we have that $r_i'^T r_j' = 1$ if $i = j$ and $r_i'^T r_j' = -\frac{1}{k-1}$ otherwise. Thus, the vectors $r_1', \ldots, r_k'$ correspond to the vertices of a simplex in $\mathbb{R}^n$. Then, there exists a rotation matrix $\bar{R} \in \mathbb{R}^{n \times n}$ such that $\bar{R} r_l' = r_l$ for all $i \in [k]$ and $\bar{R}^T \bar{R} = I$. Then, consider the vectors $v_i = \bar{R} v_i'$ for $i \in [n]$. Since rotation matrices preserve norm, we have that $\|v_i\|_2 = 1$ for

all $i \in [n]$. Thus, $v_1, \ldots, v_n$ lie in the feasible set of (8). Also, we have that:

$$Y \cdot C = U^T U \cdot C$$

$$= \sum_{i=1}^{n} \sum_{j=1}^{n} A_{ij} v_i^{'T} v_j + \sum_{i=1}^{n} v_i^{'T} \sum_{l=1}^{k} \hat{h}_i^{(l)} r_l'$$

$$= \sum_{i=1}^{n} \sum_{j=1}^{n} A_{ij} v_i^{'T} \bar{R}^T \bar{R} v_j + \sum_{i=1}^{n} v_i^{'T} \bar{R}^T \sum_{l=1}^{k} \hat{h}_i^{(l)} \bar{R} r_l' \qquad (\text{since } \bar{R}^T \bar{R} = I)$$

$$= \sum_{i=1}^{n} \sum_{j=1}^{n} A_{ij} (\bar{R} v_i')^T \bar{R} v_j + \sum_{i=1}^{n} (\bar{R} v_i')^T \sum_{l=1}^{k} \hat{h}_i^{(l)} r_l \qquad (\text{since } \bar{R} r_i' = r_i)$$

$$= \sum_{i=1}^{n} \sum_{j=1}^{n} A_{ij} v_i^T v_j + \sum_{i=1}^{n} v_i^T \sum_{l=1}^{k} \hat{h}_i^{(l)} r_l.$$

Thus, corresponding to any $Y$ in the feasible set of (9), we have found vectors $v_1, \ldots, v_n$ in the feasible set of (8) such that criterion values match.

Consequently, we have shown the range of criterion values in both optimization problems is the same, and hence the optimization problems have equivalent optimal solutions.

# B  Derivation of (11)

Let $z_1, \ldots, z_n \in \mathbb{R}^d$ such that $d = m \cdot k, m \in \mathbb{Z}$, and let $C = \frac{k}{k-1}\left(I_d - \frac{1}{k}\left(1_{k \times k} \otimes I_m\right)\right)$ where $1_{k \times k}$ is a matrix filled with 1s. Let $C = S^T S$ denote the Cholesky decomposition of $C$. Further, let us segment each $z_i$ into $k$ blocks such that $z_i^b \in \mathbb{R}^m$ denotes the $b^{th}$ block. Then, we state (11) again:

$$\max_{z_i \in \mathbb{R}^d \ \forall i \in [n]} \sum_{i=1}^{n}\sum_{j=1}^{n} A_{ij} v_i^T v_j + \sum_{i=1}^{n} v_i^T \sum_{l=1}^{k} \hat{h}_i^{(l)} r_l$$

$$\text{subject to} \quad z_i \geq 0, \quad \left\|\sum_{b=1}^{k} z_i^b\right\|_2^2 = 1, \quad v_i = S z_i \ \ \forall i \in [n]. \tag{11}$$

First, we show that with the parameterization above, $1 \geq v_i^T v_j \geq \frac{-1}{k-1}$, i.e. $v_i, v_j$ satisfy the pairwise constraints. Note that with the structure of $C$ as defined, we have that

$$v_i^T v_j = z_i^T S^T S z_j$$
$$= z_i^T C z_j$$
$$= \frac{k}{k-1}\left[z_i^T z_j - \frac{1}{k}\left(\sum_{b=1}^{k} z_i^b\right)^T \left(\sum_{b=1}^{k} z_j^b\right)\right].$$

Now, since $z_i \geq 0$ and since $\left\|\sum_{b=1}^{k} z_i^b\right\|_2^2 = 1$, we have that $\|z_i\|_2 \leq 1$. Thus, by the Cauchy-Schwartz inequality, we have that $0 \leq z_i^T z_j \leq 1$, and also that, $0 \leq \left(\sum_{b=1}^{k} z_i^b\right)^T \left(\sum_{b=1}^{k} z_j^b\right) \leq 1$. Thus, we have $v_i^T v_j \geq \frac{k}{k-1}\left(0 - \frac{1}{k}\right) = -\frac{1}{k-1}$. Further, note that

$$\|v_i\|_2^2 = \frac{k}{k-1}\left(\|z_i\|_2^2 - \frac{1}{k}\left\|\sum_{b=1}^{k} z_i^b\right\|_2^2\right)$$
$$= \frac{k}{k-1}\left(\|z_i\|_2^2 - \frac{1}{k}\right) \leq 1.$$

Thus, we have that $v_i^T v_j \leq \|v_i\|_2 \|v_j\|_2 \leq 1$, establishing both bounds. Next, note that we can set the appropriate $z_i^b$ in each $z_i$ to $e_1 \in \mathbb{R}^m$ (where $e_1$ is the first basis vector), and set all the other $z_i^{b'}$ to 0, and this allows $v_i = S z_i$ to be the required vector $r_l$ on the simplex corresponding to the optimal solution to the discrete problem (7). Thus, we have that the optimal solution to (11) is at least as large as $f^\star_{discrete}$.

Our goal then is to obtain candidates $z_1, \ldots, z_n$, such that the objective is at least as large as $f^\star_{discrete}$. Additionally, if we can further guarantee that $\|z_i\|_2 = 1$ for all $i$, we are done.

Let us write the objective in (11) in terms of $z_i$. This is

$$\sum_{i=1}^{n}\sum_{j=1}^{n} A_{ij} z_i^T C z_j + \sum_{i=1}^{n} z_i^T S^T \sum_{l=1}^{k} \hat{h}_i^{(l)} r_l.$$

Let us consider the terms in the objective involving a particular $z_i$. These are

$$z_i^T \underbrace{\left(2\sum_{j \neq i}^{n} A_{ij} C z_j + S^T \sum_{l=1}^{k} \hat{h}_i^{(l)} r_l\right)}_{g_i}.$$

Note that for every index $j$ within a block in $g_i$, across the $k$ blocks, there will definitely be at least one positive entry. This is because

$$2\sum_{j\neq i}^{n} A_{ij}Cz_j + S^T\sum_{l=1}^{k}\hat{h}_i^{(l)}r_l = 2\sum_{j\neq i}^{n} A_{ij}Cz_j + S^T\sum_{l=1}^{k}\hat{h}_i^{(l)}Se_l$$

$$= 2\sum_{j\neq i}^{n} A_{ij}Cz_j + C\sum_{l=1}^{k}\hat{h}_i^{(l)}e_l$$

$$= C\underbrace{\left(2\sum_{j\neq i}^{n} A_{ij}z_j + \sum_{l=1}^{k}\hat{h}_i^{(l)}e_l\right)}_{p}$$

$$= Cp.$$

and because of the nature of the matrix $C$, the entries at a particular index $j$ across the $k$ blocks in $Cp$ will each be of the form $x - \mathrm{avg}(x)$. This fact will be useful later on.

We now consider updating each $z_i$ in a sequential manner as a block-coordinate update, just as in the original mixing method. In the following, we drop the subscript $i$ in $z_i$ and $g_i$ for convenience. Concretely, we aim to solve the problem

$$\min \quad -g^T z$$

$$\text{subject to} \quad z \geq 0; \quad \left\|\sum_{b=1}^{k} z^b\right\|_2^2 = 1. \tag{16}$$

Let us write the Lagrangian $L(z, \alpha, \lambda)$ for the above constrained optimization problem, for dual variables $\alpha \geq 0, \lambda$:

$$L(z, \alpha, \lambda) = -g^T z + \frac{\lambda}{2}\left(\left\|\sum_{b=1}^{k} z^b\right\|_2^2 - 1\right) - \alpha^T z.$$

The KKT conditions are

$$\text{Stationarity:} \quad g_i^b + \alpha_i^b = \lambda\sum_{b=1}^{k} z_i^b \quad \forall b \in [k], i \in [m]$$

$$\text{Complementary slackness:} \quad \alpha_i^b z_i^b = 0 \quad \forall b \in [k], i \in [m]$$

$$\text{Primal feasibility:} \quad z_i^b \geq 0 \quad \forall b \in [k], i \in [m]$$

$$\left\|\sum_{b=1}^{k} z^b\right\|_2^2 = 1$$

$$\text{Dual feasibility:} \quad \alpha_i^b \geq 0 \quad \forall b \in [k], i \in [m].$$

Note that now, $z_i^b$ refers to the $i^{th}$ entry in the $b^{th}$ block in $z$. Since the KKT conditions are always sufficient, if we are able to construct $z$ and $\alpha, \lambda$ that satisfy all the conditions above, $z$ and $\alpha, \lambda$ would be optimal primal and dual solutions to (16) respectively.

Towards this, let $(\cdot)_+$ denote the operation that thresholds the argument at 0, i.e.

$$(x)_+ = \begin{cases} x & \text{if } x \geq 0 \\ 0 & \text{otherwise.} \end{cases}$$

For any fixed index $i \in [m]$, let $b(i) = \arg\max_b g_i^b$ (if there are multiple, pick any). Consider the following assignment:

$$\lambda = \sqrt{\sum_{i=1}^{m} (g_i^{b(i)})_+^2}$$

$$z_i^{b(i)} = \frac{(g_i^{b(i)})_+}{\lambda}, \quad \alpha_i^{b(i)} = \begin{cases} 0 & \text{if } g_i^{b(i)} > 0 \\ -g_i^{b(i)} & \text{otherwise} \end{cases}$$

$$z_i^b = 0, \quad \alpha_i^b = -g_i^b + \lambda z_i^{b(i)} \quad \text{for } b \neq b(i).$$

Note that $\lambda > 0$, since we argued above that there will be at least one entry that will be positive across the blocks. We will now verify that this assignment satisfies all the KKT conditions. First, note that $\sum_{b=1}^{k} z_i^b = z_i^{b(i)}$. Consider stationarity: for $b(i)$, if $g_i^{b(i)} > 0$,

$$g_i^{b(i)} + \alpha_i^{b(i)} = g_i^{b(i)} = (g_i^{b(i)})_+ = \lambda z_i^{b(i)}.$$

otherwise if $g_i^{b(i)} \leq 0$, $z_i^{b(i)} = 0$ and so

$$g_i^{b(i)} + \alpha_i^{b(i)} = g_i^{b(i)} - g_i^{b(i)} = 0 = \lambda z_i^{b(i)}.$$

For $b \neq b(i)$, by construction

$$g_i^b + \alpha_i^b = \lambda z_i^{b(i)}.$$

Next, we can observe that complementary slackness holds, since either one of $z_i^b$ or $\alpha_i^b$ is always 0. Next, we verify primal feasibility. We can observe that $z_i^b \geq 0$ for all $b$. Further,

$$\left\| \sum_{b=1}^{k} z^b \right\|_2^2 = \sum_{i=1}^{m} z_i^{b(i)2} = \frac{1}{\lambda^2} \sum_{i=1}^{m} (g_i^{b(i)})_+^2 = 1.$$

Finally, we verify dual feasibility. For $b(i)$, we have that

$$\alpha_i^{b(i)} = \begin{cases} 0 & \text{if } g_i^{b(i)} > 0 \\ -g_i^{b(i)} & \text{otherwise.} \end{cases}.$$

Either way, $\alpha_i^{b(i)} \geq 0$. For $b \neq b(i)$,

$$\alpha_i^b = -g_i^b + \lambda z_i^{b(i)} = -g_i^b + (g_i^{b(i)})_+ \geq 0.$$

Thus, we observe that the constructed $z$ and $\alpha, \lambda$ satisfy all the KKT conditions. Hence, $z$ (as constructed as above) is the optimal solution to (16). Algorithm 3 precisely updates each $z_i$ based on this constructed solution. The hope at the convergence of this routine is that we will have ended up with a solution $v_1, \ldots, v_n$ such that $f(v_1, \ldots, v_n) > f_{discrete}^\star$. Empirically, we always observe that this is the case. In fact, the solution at convergence is within 5% of the true optimal solution of (11) itself. Thus, the approximation guarantees of Frieze et al. [10] go through for the rounded solution on $v_1, \ldots, v_n$ at convergence, assuming that the entries in $A$ are positive.

## C   Proof of Theorem 1

We have that

$$
\mathbb{E}[\hat{Z}] = \mathbb{E}_{X_{p_v}}\left[\mathbb{E}_{\cdot|X_{p_v}}[\hat{Z}]\right]
$$

$$
= \mathbb{E}_{X_{p_v}}\left[\mathbb{E}_{\cdot|X_{p_v}}\left[\sum_{x\in X_{p_v}}\exp(f(x)) + \frac{1}{R}\sum_{x\in X_\Omega}\frac{\exp(f(x))}{q}\right]\right]
$$

$$
= \mathbb{E}_{X_{p_v}}\left[\sum_{x\in X_{p_v}}\exp(f(x)) + \frac{1}{R}\mathbb{E}_{\cdot|X_{p_v}}\left[\sum_{x\in X_\Omega}\frac{\exp(f(x))}{q}\right]\right]
$$

$$
= \mathbb{E}_{X_{p_v}}\left[\sum_{x\in X_{p_v}}\exp(f(x)) + \frac{1}{Rq}\sum_{x\in X_\Omega}\mathbb{E}_{\cdot|X_{p_v}}[\exp(f(x))]\right]
$$

$$
= \mathbb{E}_{X_{p_v}}\left[\sum_{x\in X_{p_v}}\exp(f(x)) + \frac{1}{Rq}\sum_{x\in X_\Omega}\sum_{y\in\{[k]^n\setminus X_{p_v}\}}q\cdot\exp(f(y))\right]
$$

$$
= \mathbb{E}_{X_{p_v}}\left[\sum_{x\in X_{p_v}}\exp(f(x)) + \frac{1}{R}\sum_{x\in X_\Omega}\sum_{y\in\{[k]^n\setminus X_{p_v}\}}\exp(f(y))\right]
$$

$$
= \mathbb{E}_{X_{p_v}}\left[\sum_{x\in X_{p_v}}\exp(f(x)) + \frac{1}{R}\cdot R\cdot\sum_{y\in\{[k]^n\setminus X_{p_v}\}}\exp(f(y))\right]
$$

$$
= \mathbb{E}_{X_{p_v}}\left[\sum_{x\in X_{p_v}}\exp(f(x)) + \sum_{y\in\{[k]^n\setminus X_{p_v}\}}\exp(f(y))\right]
$$

$$
= \mathbb{E}_{X_{p_v}}\left[\sum_{x\in[k]^n}\exp(f(x))\right]
$$

$$
= \mathbb{E}_{X_{p_v}}[Z]
$$

$$
= Z.
$$

Thus, the estimate $\hat{Z}$ given by Algorithm 4 is unbiased.

# D   Pseudocode for AIS

Our implementation of AIS has 3 main parameters: the number of temperatures in the annealing chain (denoted $K$), the number of cycles of Gibbs sampling while transitioning from one temperature to another (denoted $num\_cycles$), and the number of samples used (denoted $num\_samples$). First, we define $K + 1$ coefficients $0 = \beta_0 < \beta_1 < \cdots < \beta_K = 1$ . Then, given a general $k$-class MRF problem instance as defined in Sections 3, 4, let

$$f(x) = \sum_{i=1}^{n} \sum_{j=1}^{n} A_{ij} \hat{\delta}(x_i, x_j) + \sum_{i=1}^{n} \sum_{l=1}^{k} \hat{h}_i^{(l)} \hat{\delta}(x_i, l).$$

Further, define functions $f_k$ as follows:

$$f_k(x) = \left( \frac{1}{k^n} \right)^{1 - \beta_k} \left( \exp(f(x)) \right)^{\beta_k} .$$

Also, let $p_0$ denote the uniform distribution on the discrete hypercube $[k]^n$. The complete pseudocode for our implementation of AIS is then provided below:

---

**Algorithm 5** Annealed Importance Sampling

---

1: **procedure** GIBBSSAMPLING($x, \beta_k, num\_cycles$)
2:     Let $p(x) \propto \left( \exp(f(x)) \right)^{\beta_k}$
3:     **for** $cycle = 1, 2 \ldots, num\_cycles$ **do**
4:         **for** $i = 1, 2, \ldots, n$ **do**
5:             $x_i \leftarrow$ Sample $p(x_i | x_{-i})$
6:         **end for**
7:     **end for**
8:     **return** $x$
9: **end procedure**

10: **procedure** AIS($K, num\_cycles, num\_samples$)
11:     **for** $i = 1, 2 \ldots, num\_samples$ **do**
12:         Sample $x \sim p_0$
13:         $w^{(i)} \leftarrow 1$
14:         **for** $k = 1, 2, \ldots, K$ **do**
15:             $w^{(i)} \leftarrow w^{(i)} \cdot \frac{f_k(x)}{f_{k-1}(x)}$
16:             $x \leftarrow$ GIBBSSAMPLING($x, \beta_k, num\_cycles$)
17:         **end for**
18:     **end for**
19:     **return** $Z = \frac{1}{num\_samples} \sum_{i=1}^{num\_samples} w^{(i)}$
20: **end procedure**

---

# E    Mode estimation comparisons

Here, we compare the mode estimates given by $M^4$ and $M^4+$ with max-product belief propagation and decimation algorithm given in libDAI [22] over complete graphs across a range of coupling strengths for $k = 2, 3, 4, 5$.

(a) $k = 2, n = 20$    (b) $k = 3, n = 10$    (c) $k = 4, n = 8$    (d) $k = 5, n = 7$

Figure 4: Mode estimation comparison with max-product BP and decimation

We can observe that for both methods, the relative errors are very small ($\sim 0.018$ at worst) compared to the other methods, but $M^4+$ suffers a little for larger $k$.

Next, we show the results for the mode estimation task (timing comparison versus AIS) on complete graphs for $k = 2, 3, 4, 5$. The coupling matrices are fixed to have a coupling strength $CS(A) = 2.5$.

(a) $k = 2, n = 20$    (b) $k = 3, n = 10$    (c) $k = 4, n = 8$    (d) $k = 5, n = 7$

Figure 5: Mode estimation comparison with AIS

We can observe that both $M^4$ and $M^4+$ are able to achieve an accurate estimate of the mode much quicker than AIS across different values of $k$.

# F   Performance of AIS with varying parameters

Here, we demonstrate how the performance of AIS is affected on separately varying the parameters $K$ and $num\_cycles$ (Algorithm 5) in the partition function task. We consider similar problem instances described in Section 5 in the paper:

1. We fix $num\_cycles = 1$ and vary $K$. Figure 6 shows the results. We can observe that increasing $K$ helps increase the accuracy of the estimate of $Z$, but also becomes very expensive w.r.t. time.

(a) Complete graph $k = 2, n = 20$      (b) ER graph $k = 2, n = 20$      (c) Complete graph $k = 3, n = 10$

Figure 6: Variation of $K$ in AIS

2. Next, we fix $K$ and vary $num\_cycles$ in the Gibbs sampling step. Figure 7 shows the results. We can observe that increasing $num\_cycles$ helps increase the accuracy of the estimate of $Z$ (although the effect is much less pronounced when compared to increasing $K$), but also becomes very expensive w.r.t. time.

(a) Complete graph $k = 2, n = 20$      (b) ER graph $k = 2, n = 20$      (c) Complete graph $k = 3, n = 10$

Figure 7: Variation of $num\_cycles$ in AIS

# G   Image Segmentation - more results

We describe in more detail the setting in DenseCRF [19]. Let $f_i$ denote the feature vector associated with the $i^{th}$ pixel in an image e.g. position, RGB values, etc. Then, the image segmentation task is to compute the configuration of labels $x \in [k]^n$ for the pixels in an image that maximizes:

$$\max_{x \in [k]^n} \sum_{i<j} \mu(x_i, x_j) \bar{K}(f_i, f_j) + \sum_i \psi_u(x_i).$$

The first term provides pairwise potentials where $\bar{K}(f_i, f_j)$ is modelled as a Gaussian kernel consisting of smoothness and appearance kernels and the coefficient $\mu$ is the label compatibility function. The second term corresponds to unary potentials for the individual pixels. In keeping with the SDP relaxation described above, we relax each pixel to $\mathbb{R}^d$ to derive the following optimization problem:

$$\max_{v_i \in \mathbb{R}^d, \; \|v_i\|_2 = 1 \; \forall i \in [n]} \sum_{i<j} \bar{K}(f_i, f_j) v_i^T v_j + \theta \sum_{i=1}^n \sum_{l=1}^k \log p_{i,l} \cdot v_i^T r_l. \tag{15}$$

In the first term above, the term $v_i^T v_j$ models the label compatibility function $\mu$, and we can observe that if $\bar{K}(f_i, f_j)$ is large i.e. the pixels are similar, it encourages the vectors $v_i$ and $v_j$ to be aligned. The second term models unary potentials $\phi_u$ from available rough annotations, so that we have a bias vector $r_l$ for each label, and the term $\log p_{i,l}$ plugs in our prior belief based on annotations of the $i^{th}$ pixel being assigned the $l^{th}$ label. The coefficient $\theta$ helps control the relative weight on the pairwise and unary potentials. The mixing method update for the above objective is:

$$v_i \leftarrow normalize \underbrace{\left( \sum_{j \neq i} \bar{K}(f_i, f_j) v_j + \theta \sum_{l=1}^L \log p_{i,l} \cdot r_l \right)}_{G_i}. \tag{17}$$

We note here that computing the pairwise kernels $\bar{K}(f_i, f_j)$ naively has a quadratic time complexity in $n$, and for standard images, the number of pixels is pretty large, making this computation very slow. Here, we use the high-dimensional filtering method as in DenseCRF [19] which provides a linear time approximation for simultaneously updating all the $v_i$s as given by the update in (17). However, because of the simultaneous nature of the updates, we are no longer employing true coordinate descent. Hence, we instead propose to use a form of gradient descent with a small learning rate $\alpha$ to update each of the $v_i$s as follows. Here, the $G_i$s are those that are simultaneously given for all $i$ at once by the high-dimensional filtering method:

$$v_i \leftarrow normalize(v_i + \alpha \cdot G_i).$$

At convergence, we use the same rounding scheme described in Algorithm 2 above to obtain a configuration of labels for each pixel. Figures 8, 9 below show the results of using our method for performing image segmentation on some benchmark images obtained from the works of DenseCRF [19], Lin et al. [20]. We can see that our method produces accurate segmentations, competitive with the quality of segmentations demonstrated in DenseCRF[19].

The naive runtime for segmenting a standard (say 400x400) image by our method (without any GPU parallelization) is roughly $\sim 2$ minutes. We remark here that performing each segmentation constitutes randomly initializing the $v_i$ vectors and solving (15) via the mixing method, and also performing a few rounds of rounding. However, with parallelization and several optimizations, we believe that there is massive scope for significantly reducing this runtime.

Figure 8: Original image, annotated image, segmented image

Figure 9: Original image, annotated image, segmented image