[Reviews · NeurIPS 2020]

Review 1

Summary and Contributions: The authors propose a novel linear-sized SDP for MAP inference in Potts models that the also extend to a sampling strategy in the low temperature limit. Experimental results on synthetic and real data sets are used to validate the proposed scheme.

Strengths: The SDP construction appears novel to me and the problem of MAP estimation in such models is of significant interest.

Weaknesses: - Experiments don't really compare against anything but AIS (which is a reasonable baseline). There are lots of other (approximate) inference techniques that should be compared against. - For image segmentation, it might be good to place this in the context of the state-of-the-art for this task (even though, of course the goals isn't necessarily to beat this). - The partition function approximation scheme only really makes sense to me very close to the zero temperature limit. I don't think that this approach would really be super interesting beyond that. For comparison, how does importance sampling with say a naive mean field (fit) + plus uniform as a proposal do for these problems. I'd be surprised if it is too different. Again, I think it is important to have more baselines here.

Correctness: The presented results appear to be correct, though I did not check all of the details in the supplementary material.

Clarity: The paper is easy to read but is a bit too colloquial in parts.

Relation to Prior Work: The paper is well-positioned with respect to the SDP approximations of interest, but could probably use some additional references on approximate inference in Potts models.

Reproducibility: Yes

Additional Feedback: With more detailed experimental results, I would be happier to increase my score. The experiments are, in my opinion, a little on the low end for NeurIPS.


Review 2

Summary and Contributions: The authors propose an approximation to the MAP state and the partition function of k-state undirected models, based on SDP relaxation and importance sampling. To this end, a new alternate relaxation to the MAP problem is proposed which reduces the number of constraints to be linear in the number of variables.

Strengths: + Accurate MAP inference and partition function approximation belong to the hardest and most important problems in ML and AI + The formal presentation of the proposed method is very good and it is easy to follow the authors reasoning + Experiments show that the proposed method delivers very good quality and is faster than competing methods

Weaknesses: - Theoretical guarantees which are provided for the approximate MAP-state and the partition function are very weak (compared to other state-of-the-art methods) - Some important related work has not been cited - The core contribution relies mostly on existing ideas First, the proposed approach for solving the MAP problem is novel an very interesting. However, the last step employs a randomized rounding procedure with unknown quality. It should be possible to prove something here, but such a result is not provided in the paper. Other works in this field provide much nicer insights into the problem structure. See, e.g., David A. Sontag, Talya Meltzer, Amir Globerson, Tommi S. Jaakkola, Yair Weiss: Tightening LP Relaxations for MAP using Message Passing. UAI 2008: 503-510 Second, the proposed approach for approximating the partition function guarantees that the estimate is asymptotically unbiased. While this is nice, even the simplest approximation methods to the partition function deliver unbiased estimates. State-of-the-art methods require bounds on the variance or the approximation error. See, e.g., Stefano Ermon, Carla P. Gomes, Ashish Sabharwal, Bart Selman: Taming the Curse of Dimensionality: Discrete Integration by Hashing and Optimization. ICML (2) 2013: 334-342

Correctness: All results are theoretically sound and are carefully derived. The same holds for the experimental evaluation.

Clarity: The paper is very well written and it is easy to follow the authors reasoning.

Relation to Prior Work: Some important prior work on discrete integration by hashing and quadrature based methods are not cited, but the authors asserted to include them in the camera ready version.

Reproducibility: Yes

Additional Feedback: The authors frequently mention a "general k-state MRF". But in fact, a "general" MRF usually means that high-order factors are supported. The authors shall use the term "pairwise k-state MRF" instead, to make clear that their method does not support high-order factors.


Review 3

Summary and Contributions: The paper presents a new semidefinite programming relaxation that can be used for inference with a binary multiclass (potts-model) MRFs. The approach is based on the classical k-way max cut approximation algorithm that uses SDP relaxation and randomized rounding. The algorithm presented here uses a similar randomized rounding procedure but gives an alternative convex relaxation that is more efficient when the number of variables is higher than the number of values/labels/states.

Strengths: The new SDP relaxation is useful because it only uses O(n) constraints where n is the number of random variables. In contrast the standard SDP relaxation of max k-way cut uses O(n^2) constraints. Therefore the new SDP relaxation can be much more efficient if n is much larger than k, such as in image segmentation.

Weaknesses: It does not appear that the theoretical guarantees of the classical k-way max cut algorithm translate to this new SDP relaxation. The new SDP relaxation appears to be weaker than the original one. Although the discrete optimization problem (5) is identical to the discrete optimization problem (8), the relaxation of (5) is not identical to the relaxation of (8). It appears that a feasible solution to the relaxation of (5) leads to a feasible solution to the relaxation of (8), but not vice versa, so the original k-way SDP relaxation is tighter than the new one. The importance sampling approach is a heuristic and the theoretical results (unbiased estimation for the partition function) are not surprising since this can be done with a generic proposal distribution. The mixing of the randomized rounding with uniform sampling is an ad-hoc heuristic that points to shortcomings of the approach. The empirical results are anecdotal and there does not appear to be discussion of running time in the image segmentation experiments. I am not convinced the method is useful in practice -- together with the lack of strong theoretical results I find the approach promising but not convincing.

Correctness: The specific claims appear to be correct, except for in the intro in line 26 where the authors incorrectly state that an exponentially large number of configurations imply hardness of optimization. Obviously this is wrong as optimization can be done for example for tree-structured MRFs via dynamic programming, etc. Similarly the authors claims that computing Z is #P complete *because* there is an exponentially number of terms (line 31). Although the problem is #P complete this does not follow simply from the number of terms to be summed. (see for example the matrix tree theorem, etc). So the authors are sloppy with their complexity theory.

Clarity: generally yes but I found discussion of "theoretical guarantees" to be misleading. Some discussion the paper needs to be clarified. For example, the authors point out the classical k-way max cut approximation method has strong theoretical guarantees on the quality of the approximation but do not point out their new relaxation does not have the same approximation guarantees.

Relation to Prior Work: yes

Reproducibility: Yes

Additional Feedback: I have read the authors response and that did not change my assessment. I think this is a fine piece of work but the paper needs to be much more clear about the theoretical guarantees of the approach.

[Author Response · NeurIPS 2020]

We thank the reviewers for their thoughtful and constructive feedback, as well as the pointers to related work, which we
plan on incorporating in the next revision.

**R1** *Comparisons to baselines:* We acknowledge that there is indeed a vast literature of approximate inference methods
to compare against. However, as we pointed out in our experiments section, we would like to mention again that Park et
al. [24] have provided comparisons and shown that they significantly outperform popular techniques like mean-field
approximation, belief propagation, etc. on the binary partition function tasks considered in the paper, which was why
we skipped those. Further, the variations in coupling strengths demonstrate that our method works in a variety of
temperature settings. For the general $k$-class MRF, since AIS is a strong sampling-based technique, we focused on
running it with different parameter settings to verify the superior performance of our method. However, we would be
happy to include other relevant baselines in the general case (e.g. RAISE) in the final version. In our image segmentation
experiments, the goal was to demonstrate the capability of our model to actually scale up to real-world MRFs with
thousands of nodes, whilst also delivering good results (since the standard $k$-class relaxation with quadratic constraints
would quickly not scale). To put this in context with modern state-of-the-art methods, we provide here a qualitative
comparison with a trained UNet (Ronneberger et al. 2015) Figure 1a on the Carvana Image Masking Challenge. As can
be observed, the segmentations provided by our method are highly comparable (if not better) to those by UNet.

**R2** *Lack of theoretical guarantees and missing references:* We admit the lack of theoretical guarantees for the general
$k$-class case of our algorithm. We remark here that since we are indeed solving a different relaxation with a modified
objective and lesser constraints, the analysis of Frieze et al. [9] does not go through as is. But for the special case
of $k = 2$, we can provide approximation bounds following $\log \mathbb{E} \exp(x^T A x) \leq \max x^T A x$ together with the 0.878
bounds of Goemans-Williamson. We will add the proof for the same in the revision. We remark here that although the
classical relaxation (6) does provide theoretical guarantees, solving it becomes infeasible for large MRFs. In contrast,
from our empirical results, our relaxation *does* scale up, and we admit that we have traded off a lack of guarantees with
practical performance. As for the additional references on discrete integration by hashing pointed out, we will definitely
include it (possibly with a comparison on the binary MRF task) in the revised version.

**R3** *Tightness of our relaxation:* We acknowledge that our relaxation (11) is indeed looser than the classical $k$- class
relaxation (6). However, due to the quadratic number of pairwise constraints in (6), we remark here again that solving
this SDP quickly becomes practically infeasible with increase in the MRF size $n$. The computational cost to solve this
SDP with $\sim n^2$ constraints with a traditional solver would be $O(n^6)$ [1]. Thus, our relaxation, albeit being looser, *does*
*scale up* (as seen in our image segmentation section) to practically large MRFs. Further, we empirically observe that the
mode estimates obtained *after randomized rounding* are largely the same on solving the relaxation (6) both with and
without the pairwise constraints. This is verified in Figure 1b, where we compute mode estimates on 5-class MRFs by
solving (6) both with and without pairwise constraints. Performing randomized rounding on the solutions to both these
SDPs deliver mode estimates of practically the same quality across various coupling strengths. Thus, while hugely
benefitting in runtime, our relaxation doesn't suffer much at all in quality of solution.

*Runtime and complexity-theoretic claims:* The naive runtime for segmenting a standard (say 400x400) image by our
method (without any GPU parallelization) is roughly $\sim 2$ minutes. We remark here that performing each segmentation
constitutes randomly initializing the $v_i$ vectors and solving (6) via the mixing method as in Algorithm 1, and also
performing a few rounds of rounding as in Algorithm 2. However, with parallelization and several optimizations,
we believe that there is massive scope for significantly reducing this runtime. We reiterate here that our goal in the
segmentation section was to simply demonstrate the capability of our method to scale up. As for the sloppiness in the
complexity-theoretic statements in the paper (e.g. naively equating exponential configurations to #P hardness), we will
definitely correct them in the final version.

(a) Comparison with UNet

(b) Mode estimates w/ and w/o pairwise constraints

## Footnotes

[1] See Sec 6.3.5, Pg. 72 in https://docs.mosek.com/MOSEKModelingCookbook-letter.pdf


[Meta-Review · NeurIPS 2020]

The paper has been considered reasonable, even though there have been some concerns with the clarity/exaggeration of some claims that were made (as also acknowledged in the response). This should definitely be properly addressed by the authors. There were also concerns about completeness of literature and experiments, and again the authors have proposed fixes to that (admittedly we cannot check if they are implemented, but we may choose to trust it). Since those are implementable changes, authors are expected to achieve them in great part. One related paper which was not mentioned (besides those mentioned in the reviews) is Nico Piatkowski, Katharina Morik: Stochastic Discrete Clenshaw-Curtis Quadrature. ICML 2016: 3000-3009, where they derive bounds on the approximation error for the partition function. It seems highly relevant for this paper. It is also expected that the authors will fix any sloppy situation as pointed out.